# ZIC-cHILIC Functionalized Magnetic Nanoparticle for Rapid and Sensitive Glycopeptide Enrichment from <1 µL Serum

**DOI:** 10.3390/nano11092159

**Published:** 2021-08-24

**Authors:** Tiara Pradita, Yi-Ju Chen, Elias Gizaw Mernie, Sharine Noelle Bendulo, Yu-Ju Chen

**Affiliations:** 1Institute of Chemistry, Academia Sinica, Taipei 115, Taiwan; tpradita01@gate.sinica.edu.tw (T.P.); tp7249@gate.sinica.edu.tw (Y.-J.C.); eliasgizaw6@gmail.com (E.G.M.); sobendulo@gmail.com (S.N.B.); 2Sustainable Chemical Science and Technology, Taiwan International Graduate Program, Academia Sinica, Taipei 115, Taiwan; 3Department of Applied Chemistry, National Yang Ming Chiao Tung University, Hsinchu 300, Taiwan; 4Department of Chemistry, National Taiwan University, Taipei 106, Taiwan

**Keywords:** n-dodecylphosphocholine, magnetic nanoparticle, glycopeptide, hepatocellular carcinoma, hepatitis B virus

## Abstract

Due to their unique glycan composition and linkage, protein glycosylation plays significant roles in cellular function and is associated with various diseases. For comprehensive characterization of their extreme structural complexity occurring in >50% of human proteins, time-consuming multi-step enrichment of glycopeptides is required. Here we report zwitterionic n-dodecylphosphocholine-functionalized magnetic nanoparticles (ZIC-cHILIC@MNPs) as a highly efficient affinity nanoprobe for large-scale enrichment of glycopeptides. We demonstrate that ZIC-cHILIC@MNPs possess excellent affinity, with 80–91% specificity for glycopeptide enrichment, especially for sialylated glycopeptide (90%) from biofluid specimens. This strategy provides rapidity (~10 min) and high sensitivity (<1 μL serum) for the whole enrichment process in patient serum, likely due to the rapid separation using magnetic nanoparticles, fast reaction, and high performance of the affinity nanoprobe at nanoscale. Using this strategy, we achieved personalized profiles of patients with hepatitis B virus (HBV, *n* = 3) and hepatocellular carcinoma (HCC, *n* = 3) at the depth of >3000 glycopeptides, especially for the large-scale identification of under-explored sialylated glycopeptides. The glycoproteomics atlas also revealed the differential pattern of sialylated glycopeptides between HBV and HCC groups. The ZIC-cHILIC@MNPs could be a generic tool for advancing the glycoproteome analysis, and contribute to the screening of glycoprotein biomarkers.

## 1. Introduction

Protein glycosylation is a ubiquitous post-translational modification that is estimated to occur in over 50% of the human proteome [1]. Due to their unique structure with heterogeneous glycan composition, glycan linkage, and microheterogeneity in the extracellular space, glycosylation plays major roles in many biological activities and cell processes such as protein folding, signal transduction, and cell–cell communication [2,3,4]. Aberrant glycosylation has been discovered to significantly relate to various diseases such as lung cancer [5], liver cancer [6], HIV [7], prostate cancer [8], skin cancer [9], and ovarian cancer [10]. It is among the most complex protein modifications, consisting of chain-like structures of monosaccharides (i.e., glycan) that are attached to the polypeptide backbone and terminated with myriad permutations of sialylation, sulfation, and fucosylation, which not only constitute specific recognition code, but also fine-tune the binding of underlying peptide loop and domain [11]. A vast number of competing and sequentially acting glycosyltransferases and glycosidases control the structure of glycans, resulting in a diversified pool of glycans across glycosites of a given protein. This meta-heterogeneity requires high performance tools that allow precise definition of the structure details and functional association with downstream signaling events.

Mass spectrometry (MS) has been used as a prime tool for full characterization of the glycan structures, glycosylation site, and protein carrier on glycoprotein. However, due to low abundance, heterogeneity, and low detectability caused by the ionization suppression from non-glycosylated peptides, direct analysis of glycopeptides is still challenging [3,11,12]. Various glycopeptide enrichment strategies prior to MS analysis such as lectin affinity chromatography [13,14,15], hydrazide chemistry [16,17,18], boronic acid [19,20,21,22], and hydrophilic interaction chromatography [23,24,25,26] have been widely studied. Among the glycopeptide enrichment tools, the hydrophilic interaction chromatography (HILIC) is a popular method. In HILIC, separation is mainly based on the hydrophilic interaction. Analytes are partitioned between the hydration layer surrounding the stationary phase and the non-polar component of the mobile phase. Separation can be also due to weak electrostatic interactions between the stationary phase and the analytes. Glycopeptides are significantly more hydrophilic and have several hydroxyl groups from the glycan that can participate in H-bonding. Thus, they are preferentially retained over non-glycopeptides in HILIC [11]. Several HILIC materials such as cation–anion exchange [27], siloxane [28], saccharides [29], and zwitterionic (ZIC)-based [26,30,31] were widely studied and available commercially. Wohlgemut et al. utilized ZIC-HILIC on a monolithic nano-LC column for glycopeptide enrichment, demonstrating sensitive results at single glycoprotein level, including 27 and 62 glycopeptides enriched in IgG1 and α-acid glycoprotein, respectively [23]. Jiang et al. developed a multi-parallel enrichment strategy assisted by a filter (C18 and ZIC-HILIC)-coated 96-well plate for glycopeptide enrichment, with more than 75% specificity, and 5466 N-glycosites from 2383 glycoproteins were identified from an HCC cell line [32]. The above works show significant development in the utilization of zwitterionic based HILIC for glycopeptide enrichment.

Phosphocholine is a relatively new zwitterion used in the HILIC approach (ZIC-cHILIC), and has the opposite charge arrangement compared to the commercially available ZIC-HILIC (sulfobetaine). Because the positive charge is located on the outermost part of the particle, ZIC-cHILIC has more accessible positive charge to enhance the electrostatic attraction of negatively charged glycopeptides, especially towards negatively charged sialylated glycopeptide [33,34]. Wang et al. studied the physicochemical and chromatographic properties of phosphocholine-functionalized monolith columns; a total of 24 glycopeptides, including 19 fucosylated and 2 sialyl-glycopeptide, were enriched from human IgG, suggesting a promising material for N-glycopeptide enrichment [33]. Although a vast number of glycoproteomic workflows have been developed, many suffer from the time-consuming and complicated enrichment process. Therefore, it is necessary to develop an efficient enrichment strategy toward comprehensive glycopeptide analysis.

Magnetic nanoparticles (MNPs) have been used in many applications in the fields of biomedicine, bioimaging, and bioseparation. MNPs are highly favored as solid support materials due to their large surface area, ease of operation, and high binding capacity [35]. Several studies have incorporated magnetic nanoparticles as the solid support for HILIC method [36,37,38]. Yeh et al. fabricated magnetic bead-based ZIC-HILIC material for glycopeptide enrichment. The pioneering study identified 85 N-glycosites in 53 glycoproteins from urine samples [38]. Jiao et al. synthesized ultrathin Au nanowire-assisted zwitterionic hydrophilic magnetic graphene oxide (GO-Fe_3_O_4_/SiO_2_/AuNWS/L-Cys) for glycopeptide enrichment, achieving identification of 793 glycopeptides from 467 glycoproteins in mouse liver [39]. Based on large surface-to-volume ratio, high ligand affinity, and rapid separation of MNPs, we developed nanoprobe-based affinity mass spectrometry as a rapid immunoassay with high sensitivity [40]. Further implementation has extended the development of one-pot two-nanoprobe assay, with 10-fold superior glycopeptide signals of serum biomarker [41], and glycotope-specific enrichment and identification of glycopeptides from cell line [15]. Among the above works, however, coverage of sialylated glycopeptides remains to be improved.

Taking advantage of rapid and sensitive enrichment of magnetic nanoparticles (MNPs) for glycopeptides, in this study, we designed and fabricated ZIC-cHILIC functionalized magnetic nanoparticles (ZIC-cHILIC@MNPs) and developed a robust workflow for glycopeptide enrichment from low input sample, especially for sialylated glycopeptides. The ZIC-cHILIC material (the phosphocholine functional group), was conjugated on the MNPs (Figure 1A). The MNPs were expected to possess several advantages as an enrichment probe: increasing the phosphocholine density, providing a fast separation process, and requiring low amounts of sample. ZIC-cHILIC contains zwitterionic phosphocholine functional groups and provides hydrophilic interaction with glycopeptides on the surface (Figure 1B). The material has a positively charged choline group externally exposed, which can be more accessible and increases electrostatic attraction for negatively charged glycopeptides, especially sialylated glycopeptides. In addition, the immobilized hydration layer favors the hydrophilic analytes, whilst the hydrophobic analytes are retained in the mobile organic layer [31]. The ZIC-cHILIC@MNPs were expected to combine HILIC mechanism with zwitterionic features to improve the selectivity towards hydrophilic polar molecules (i.e., glycan moiety of glycopeptides).

Using the ZIC-cHILIC@MNPs, we then developed a simple and robust strategy to directly enrich intact glycopeptides of targeted glycoprotein or serum samples (Figure 1C). Integrating the advantages of MNPs (large ligand density, ease of operation, and high binding capacity [15,35]), the MNPs-based enrichment achieved fast separation within 2–5 min. Alagesan et al. reported that acetonitrile (ACN) resulted in the least non-specific absorption of free peptides in the ZIC-HILIC approach [26]. Myslinc et al. discovered that Trifluoroacetic acid (TFA) can reduce the overlapping hydrophilicity between glycopeptides and non-glycosylated peptides [24]. Presumably, using TFA for ion pairing will decrease the non-specific binding of free peptides during glycopeptide enrichment. Thus, we also proposed the use of ACN as the organic solvent and TFA as ion-pairing reagent. Utilizing standard glycoprotein digests, we demonstrated high sensitivity on the limit of detection from 2 μg down to 10 ng of standard HRP and fetuin. We further applied this strategy to explore the serum glycoproteome profile of individual HCC and HBV patients by using only 0.15–2 μL of serum. Our results demonstrated that ZIC-cHILIC@MNPs provide a high enrichment specificity of 81–90% for sialylated glycopeptides in sera.

## 2. Materials and Methods

### 2.1. Materials and Chemicals

Iron(III) acetylacetonate (≥99%), absolute ethanol, n-hexane, toluene, acetonitrile, and C18 ziptips were purchased from Merck (Darmstadt, Germany). Standard proteins (fetuin, horseradish peroxidase (HRP)), phosphate buffer saline (PBS), acetic acid (AA), formic acid (FA), α-cyano-4-hydroxycinnamic acid (CHCA), ammonia (25%), 1,2-hexadecanediol (90%), (3-aminopropyl) trimethoxysilane (97%), sodium hydroxide (97%), tris(2-carboxyethyl) phosphine (≥98%), triethylammonium bicarbonate (TEABC), ninhydrin, and chloroform were obtained from Sigma Aldrich (St. Louis, MI, USA). Tetraethyl orthosilicate (≥98%), phenyl ether (99%), and oleylamine (80–90%) were purchased from Acros Organics (Fair Lawn, NJ, USA). N-dodecylphosphocholine was obtained from Anatrace (Maumee, OH USA). Trifluoroacetic acid (TFA) and ethyl acetate were purchased from Wako pure chemical industry (Osaka, Japan). Dithiothreitol (DTT) and iodoacetamide (IAM) were purchased from JT Baker (Phillipsburg, NJ, USA). Modified sequencing-grade trypsin was purchased from Promega (Madison, WI, USA). Protease inhibitor was purchased from Roche (Mannheim, Germany). SDB-XC Empore^TM^ disk membranes were purchased from 3M^TM^ (St. Paul, MN, USA).

### 2.2. Synthesis of ZIC-cHILIC@MNPs

The synthesis of Fe_3_O_4_ magnetic nanoparticles was conducted using the thermal decomposition method following previous protocol [42,43]. Two mmol of iron(III) acetylacetonate, 10 mmol of 1,2-hexadecanediol, 6 mmol of oleic acid, 6 mmol of oleylamine, and 20 mL of diphenyl ether were mixed in a three-neck round bottom flask at 200 °C for 2 h under nitrogen. The mixture was then refluxed at 300 °C for 1 h. After the reaction, the mixture was cooled down to room temperature. Ethanol was added to precipitate the nanoparticles. The mixture was centrifuged at 6000 rpm for 10 min. Then, the supernatant was removed. The magnetic nanoparticles were re-dispersed in hexane containing 0.15 mmol each of oleic acid and oleylamine. They were then re-precipitated and washed with ethanol. After centrifugation, the nanoparticles were dispersed in hexane and stored at 4 °C.

For the synthesis of ZIC-cHILIC@MNPs, 15 µg of core Fe_3_O_4_ nanoparticles dissolved in 0.5 mL chloroform and 96.66 mg of n-dodecylphosphocholine dissolved in 5 mL water were mixed for 1 h. The resulting mixture was then mixed at 60 °C for 20 min. The temperature was increased to 70 °C, and the pH was increased by adding 1 mL of 2 M aqueous sodium hydroxide and 30 mL water. Then, 250 µL of tetraethyl orthosilicate and 3 mL of ethyl acetate were added sequentially. The mixing continued for 30 min. Lastly, 90 µL of (3-aminopropyl)-trimethoxysilane was added with continuous stirring for another 4 h. After cooling to room temperature, the magnetic nanoparticles were washed three times with ethanol and vacuum dried. The synthesis of ZIC-cHILIC@MNPs is illustrated in Appendix A.

### 2.3. Characterization of ZIC-cHILIC@MNPs

The morphology of ZIC-cHILIC@MNPs was examined by UHR FE-SEM ULTRA Plus (Carl Zeiss, Jena, Germany) with a 17 kV energy source and a magnification of X. The elemental composition of ZIC-cHILIC@MNPs was identified by EDX X-Max (Oxford Instrument, Oxfordshire, UK). The surface functional groups were identified by Spectrum 100 Fourier Transformed Infrared (FTIR) spectroscopy (Perkin Elmer, Waltham, MA, USA). The absorbance was scanned from 450 to 4000 cm^−1^. ATR correction was performed and the data was processed using IR Solutions. Zeta potential (ζ) measurements were conducted using Malvern Zetasizer Nano ZS (Worcestershire, UK). Core MNPs and ZIC-cHILIC@MNPs were dispersed in hexane and water, respectively. The data was processed using Zetasizer Software 7.01.

### 2.4. Sample Reparation for Standard Glycoproteins

Two standard glycoproteins including horseradish peroxide (HRP) and fetuin were dissolved in 25 mM TEABC, and 2 μL of individual serum samples from 3 HCC and 3 HBV patients was diluted with PBS buffer. The samples were reduced using the final 5 mM of TCEP at 37 °C for 30 min and then alkylated with the final 20 mM of iodoacetamide (IAM) in the dark at 37 °C for 30 min. Additional dithiothreitol (DTT) reagent was added to quench the remaining IAM and reacted for another 10 min at room temperature. Trypsin (sample:trypsin (*w*/*w*) = 20:1) was added and incubated at 37 °C for 16 h. After digestion, all samples were desalted by SDB-XC stage tip.

### 2.5. Glycopeptide Enrichment from Standard Proteins

One hundred micrograms of ZIC-cHILIC@MNPs was suspended in 10 μL of incubation buffer (0.1/9.9/90, TFA/H_2_O/ACN, *v*/*v*/*v*) for activation, and sonicated for 2 min. The tryptic-digested peptides of fetuin and HRP were added into the ZIC-cHILIC@MNPs suspension and incubated at room temperature for 2 min. After collecting the glycopeptides-associated ZIC-cHILIC@MNPs by magnet, the supernatant was removed and washed with 10 μL of incubation buffer twice. The enriched glycopeptides were eluted from the nanoparticles with 10 μL elution buffer (0.5/4 4.5/55, FA/H_2_O/ACN, *v*/*v*/*v*) in a vortex shaker for 2 min, desalted by C18 ziptip and re-dissolved for MALDI-TOF MS analysis.

### 2.6. Glycopeptide Enrichment from Patient Serum

Participant recruitment and serum sample collection was approved and followed by the Institutional Review Board on Biomedical Science Research, Academia Sinica, and National Taiwan University Hospital (NTUH). Informed consent was obtained from all participants in this study. Two hundred micrograms of ZIC-cHILIC@MNPs was suspended in 20 μL of incubation buffer (0.1/9.9/90, TFA/H_2_O/ACN, *v*/*v*/*v*) for activation, and sonicated for 2 min. The tryptic digests of 2 μL of patient serum were added into the ZIC-cHILIC@MNPs suspension, and the mixture was incubated at room temperature for 5 min. After removing the supernatant with an external magnet, the glycopeptide-loaded magnetic nanoparticles were washed with 100 μL of incubation buffer followed by sequential washing with washing buffer (6/14/80, AA/H_2_O/ACN, *v*/*v*/*v*) twice. The enriched glycopeptides were eluted from the nanoparticles with 50 μL of elution buffer (0.5/44.5 /55, FA/H_2_O/ACN, *v*/*v*/*v*) for 5 min, desalted by SDBXC stage tip, and re-dissolved with 10 μL 0.1% FA for LC-MS/MS analysis.

### 2.7. Mass Spectrometry Analysis

The enriched glycopeptides from the digested standard HRP and fetuin were analyzed using ultrafleXtreme MALDI-TOF/TOF (Billerica, MA, USA). The instrument was equipped with a Nd-YAG laser with a wavelength of 355 nm and a fixed laser intensity of 20%. The analyses were performed in positive ion linear mode from 1000 to 7000 Da. Six thousand laser shots were averaged to obtain the spectra, followed by Gaussian smoothing and baseline correction using Bruker Daltonics flexAnalysis version 3.4 software. The glycopeptides enriched from HBV and HCC patient serum were analyzed by Thermo Scientific™ Orbitrap Fusion Lumos™ Tribrid™ mass spectrometer coupled with UltiMate™ 3000 RSLCnano System (Thermo Fisher Scientific, Bremen, Germany) equipped with a nanospray interface (Proxeon, Odense, Denmark). The samples were loaded onto a 50 cm length C18 BEH column (Waters, Milford, MA, USA) packed with 1.7 μm particles with a pore size of 130 Å and were separated with a segmented 60 or 120 min gradient with the mobile phase (Buffer A, 0.1% FA in water; Buffer B, 100 ACN with 0.1% FA) with 2–85% buffer B at 250 nL/min flow rate. Orbitrap survey MS^1^ scans of peptide precursors were acquired from m/z 400–2000 at 120 K resolution, with a target value of 2 × 10^5^ ion count. The included charge states were 2 to 6 and the maximum injection time was 100 msec. Tandem MS was performed by isolating the precursor ions at a window of 2 Th in the quadrupole and fragmented by a product-dependent stepped higher-energy collisional dissociation (HCD) workflow. For MS^2^, first, HCD at a normalized lower-collision energy setting of 26 was acquired for the top-10 precursors detected in MS^1^, and detected in the Orbitrap at a resolution setting of 30 K, with a target value of 5 × 10^4^ ion count and a maximum injection time of 25 msec. The second step was triggered when the diagnostic oxonium ions detected in the acquired HCD MSMS spectra were m/z 138.0545 for the HexNAc^+^ fragment; m/z 204.0867 for the HexNAc^+^ fragment; and/or m/z 366.1396 for the HexHexNAc^+^ fragment. The product-dependent stepped HCD fragmentation was set at 35 ± 8% collision energy and detected simultaneously in the Orbitrap at a resolution setting of 30 K with a target value of 5 × 10^4^ ion count. The maximum injection time was 50 msec and the dynamic exclusion duration was set to 40 s with 10 ppm tolerance around the selected precursor and its isotopes. Monoisotopic precursor selection was turned on.

### 2.8. Data Analysis

Raw data were processed using Proteome Discoverer 2.5 (PD2.5; ThermoFisher Scientific) and converted to .mgf files. The .mgf files were used to count the number of MS/MS spectra containing any two of the three common diagnostic oxonium ions (m/z 366.11 for HexHexNAc^+^, 204.08 for HexNAc^+^, and 138.06 for HexNAc^+^ fragments) with S/N ≥ 10 as derived from glycopeptides. Another three diagnostic oxonium ions of sialylation (m/z 274.09 for Neu5Ac^+^-H_2_O, 292.09 for Neu5Ac^+^, and 657.24 for Neu5Ac-Hex-HexNAc^+^) were also used as evidence of MS/MS spectra derived from sialylated glycopeptides. For identification of intact glycopeptides, the raw data were queried using Byonic™ (v3.6, Protein Metrics) for tryptic peptides with less than two cleavage sites, a precursor ion mass tolerance of 10 ppm, and a fragment ion tolerance of 0.05 Da for HCD spectra. Protein FASTA files, including Human (Swiss-Prot database, v2021-05-06, total 20,324 sequences from human) and 2 standard glycoproteins (bovine fetuin, P12763; HRP, P00433) were used for protein identification. The built-in human N-glycans database (with 57 human plasma and ~182 human glycans) and O-glycans database (70 human) were used for identification of glycan composition. Carbamidomethyl (C) was selected as fixed modification; deamidation (NQ) and oxidation (M) were selected as variable (common) modifications. N- and O-glycan were selected as “rare”. The maximum of total common modification was set as 4 and rare modification was set as 1. The score of identified glycopeptides was higher than 30 and confident identification of glycopeptides was set at >100; reversed peptide sequence identification was also considered with a protein FDR of 1%, or 20 reverse count. The high confidence of glycopeptide sequence was also considered the Pep2D values with FDR <0.01. The label-free quantitation was further processed by using chromatographic alignment with *m*/*z* and retention time of identified intact glycopeptides in PD2.5 software. The specificity of identified (sialylated) glycopeptides was further compared to total MS/MS spectra.

## 3. Results and Discussion

### 3.1. Characterization of ZIC-cHILIC@MNPs

The ZIC-cHILIC@MNPs were prepared by incubation of core MNPs and n-dodecylphosphocholine through comprehensive van der Waals interaction, giving ZIC-cHILIC@MNPs a zwitterionic feature where the positive charge of the choline group is exposed on the outer layer and the negative charge of phosphoryl group is on the inner part of the nanoparticle. To characterize the chemical bonding of different materials, the functional group of core MNPs and ZIC-cHILIC@MNPs were analyzed by Fourier-Transform Infrared Spectroscopy (FT-IR). As shown in Figure 2A, the peak at 3350–3010 cm^−1^ validates the N–H stretching of 3-aminopropylsiloxane on MNPs, although it slightly overlaps with the O–H (3550–3200 cm^−1^) and C–H (2933–2852 cm^−1^) peaks. In addition, a sharp peak at 1065 cm^−1^ representing the Si–O stretch from the Si surface coating of silicon dioxide was observed. The peak for P–O stretch of n-dodecylphosphocholine was barely observed in the shoulder of the major peak of 1065 cm^−1^, likely due to its overlapping with the abundant Si–O stretch (1065 cm^−1^). The decrease in peak intensity of Fe–O (655 cm^−1^) in the ZIC-cHILIC@MNPs curve indicates additional surface coating.

As shown in Figure 2B,C, the ZIC-cHILIC@MNPs showed significant increase in the atomic weight percentages of oxygen (O) and carbon (C) relative to iron (Fe). The increase in oxygen and carbon were due to functionalization with the zwitterionic n-dodecylphosphocholine and coating with silica, suggesting additional surface modification of core MNPs. However, phosphorus (P) cannot be detected due to its lower abundance than the other dominant elements (O, C, and Si). The polarity of core MNPs and ZIC-cHILIC@MNPs was also investigated (Figure 2D). Prior to functionalization, core MNPs were well-dispersed in hexane, indicating that the hydrophobic oleic acid and oleylamine were successfully coated on the core MNPs. After functionalization, the ZIC-cHILIC@MNPs were suspended in water, suggesting the modification of the polar moieties of n-dodecylphosphocholine and aminopropylsiloxane on the surface of the core MNPs. These results also revealed the change in nanoparticle dispersibility from non-aqueous to aqueous medium upon functionalization. We further measured the zeta potential (ζ) to analyze the net charge of ZIC-cHILIC@MNPs. Due to the hydrophobicity of the core MNPs, the zeta potential measurement (ζ) was conducted using hexane as the dispersant. Meanwhile, zeta potential measurement (ζ) of hydrophilic ZIC-cHILIC@MNPs was conducted using water as the dispersant. It showed that the net charge of core MNPs and ZIC-cHILIC@MNPs are −1.62 ± 20.20 and −22.20 ± 4.82, respectively (Appendix A). The negative charge on the surface of ZIC-cHILIC@MNPs is likely derived from several factors, including the abundant and very weakly acidic coating of silica layers on ZIC-cHILIC@MNPs [44,45] and the negative net charge of phosphocholine [33,34]. Alpert et al. hypothesized that the negative charge of phosphocholine functionality might occur because the level of salt in the mobile phase is insufficient to titrate the strong negative charges on the phosphate group in phosphocholine [46]. Though the negative charge of ZIC-cHILIC@MNPs may not align with our hypothesis about the electrostatic interaction during the enrichment, electrostatic interaction in HILIC-based enrichment is not the main interaction. Zwitterions are advantageous for HILIC-based enrichment, because they are doubly charged and located near each other.

Thus, they can attract more water and form a large and stable water layer surrounding the nanoparticles to enhance the hydrophilic interaction. Futhormore, in order to determine the nanoparticles’ morphology, Field Emission Scanning Electron Microscope (FE-SEM) imaging was conducted and it presented that ZIC-cHILIC@MNPs have spherical form with size estimated in the range of 100–200 nm (Figure 2E). It is noted that the ZIC-cHILIC@MNPs show good long-term stability for >6 months in 4 °C storage. Furthermore, a reusability test was conducted to determine the stability of ZIC-cHILIC@MNPs. Five rounds of enrichment were conducted. After each round, the magnetic nanoparticles were transferred to another tube and washed three times with the incubation buffer. After the first four rounds, the overall intensity and the number of enriched glycopeptides from HRP only slightly decreased (Appendix A). However, ZIC-cHILIC@MNPs were still able to enrich glycopeptide from HRP after five rounds of enrichment. Similarly, ZIC-cHILIC@MNPs also showed good stability after five rounds of glycopeptide enrichment from fetuin (Appendix A). These results suggested that ZIC-cHILIC@MNPs possess good stability.

### 3.2. ZIC-cHILIC@MNPs for Glycopeptide Enrichment from Standard Glycoproteins

In order to evaluate the ability of ZIC-cHILIC@MNPs in glycopeptide enrichment, two standard glycoproteins; horseradish peroxidase (HRP), with core fucosylated and core-xylosylated glycans, and fetuin, with sialylated glycans, were used. The ratio of ACN to TFA in HILIC-based materials is well known as an important factor for glycopeptide enrichment; it can considerably influence the hydrophobicity of non-glycosylated peptides during the HILIC enrichment process. A high ratio of ACN to water for mobile phase is preferable to apply in the ZIC-cHILIC approach. The high content of organic solvent in incubation and washing buffer are expected to separate and wash away the hydrophobic non-glycosylated peptides, which can help decrease the non-specific binding [24,26]. Most importantly, the ZIC-cHILIC@MNPs may have unique hydrophobicity compared to core MNPs or ZIC-cHILIC alone. To investigate the hydrophobicity effect, the concentration of ACN in the incubation solvent was varied from 85% to 95%.

Before enrichment, non-glycosylated peptides predominated in the mass spectrum and caused ion suppression, resulting in the absence of glycopeptides (Figure 3A). After enrichment using ZIC-cHILIC@MNPs, a maximum of 21 intact glycopeptides from three sites were significantly observed (Figure 3B) from standard fetuin using 0.1/9.9/90 TFA/H_2_O/ACN (*v*/*v*/*v*). Whilst, in the case of HRP, a maximum of 16 intact glycopeptides from eight sites were detected with strong intensities using 0.1/4.9/95 TFA/H_2_O/ACN (*v*/*v*/*v*) (Figure 3D,E). The detailed number of detected glycopeptides from different ACN contents is presented in Appendix A and detailed information about glycopeptides enriched from HRP (Appendix A) and fetuin (Appendix A) are presented in Appendix A. The result indicated that ZIC-cHILIC@MNPs harnessed the ability to enrich not only neutral glycopeptides but also sialylated glycopeptides. In terms of glycoforms, as expected, core-fucosylated glycopeptides, such as N348-N2H3F1 (*m*/*z* 3353.41), N335-N2H3F1 (*m*/*z* 3671.7), and N266/N278-N2H3F1 (*m*/*z* 4983.2), were mostly identified from HRP. On the other hand, bi- and tri-antennary sialylated glycopeptides, including N99-N4H5S1 (*m*/*z* 5584.9), N99-N4H5S1 (*m*/*z* 5469.35), and N99-N5H6S2 (*m*/*z* 6124.9), were mostly enriched from fetuin. In addition, high mannose glycan (*m*/*z* 3121.39, N156-N2H6) and tetra-antennary sialylated glycopeptide, with the number of sialic acids up to four (*m*/*z* 7190.41, N99-N6H7S5) were also identified from fetuin. These results revealed that ZIC-cHILIC@MNPs are capable of enrichment towards a diversified pool of glycoforms. Most importantly, the enrichment process from incubation with sample and elution of enriched glycopeptides only takes 10 min, which is likely due to the fast reaction of the affinity nanoprobe at nanoscale. Based on the above result, we chose to utilize 0.1/9.9/99 TFA/H_2_O/ACN (*v*/*v*/*v*) as the incubation buffer condition, expecting better enrichment performance towards sialylated glycopeptides.

In addition, the enrichment sensitivity of ZIC-cHILIC@MNPs was also explored for different amounts of fetuin and HRP (0.01 μg, 0.05 μg, 0.1 μg, 0.25 μg, 0.5 μg, 1 μg, and 2 μg). From the observed results (Figure 3E,F), ZIC-cHILIC@MNPs enrichment coupled with MALDI-TOF mass spectrometry analysis can facilitate detection sensitivity as low as 0.1 μg and 0.01 μg of standard fetuin and HRP, respectively. This result indicates a good sensitivity of ZIC-cHILIC@MNPs for low numbers of abundant samples.

### 3.3. Application of ZIC-cHILIC@MNPs for Serum Glycoproteomics Profiling of HCC and HBV Patients

To demonstrate the utility for complex biofluid, the ZIC-cHILIC@MNPs were further applied to study the glycoproteomic profile of serum from hepatocellular carcinoma (HCC)- and hepatitis B virus (HBV)-infected patients as model systems. Previous literature has reported that aberrant glycosylation is related to the pathogenesis of HCC and HBV [47,48]. Thus, identification of glycoprotein biomarkers may provide risk stratification to monitor the transformation from HBV to HCC. Here, we further optimized the amount of ZIC-cHILIC@MNPs (100 μg to 800 μg) and the volume (1–3 μL) of one HCC serum to achieve optimal performance in glycoproteomics coverage under 60 min LC-MS/MS gradient. By counting the number of MS/MS spectra containing sialylated oxonium ions, which are signature ions of sialylated glycopeptides, compared to total MSMS spectra, high enrichment specificity (88–92%) of sialylated glycopeptides was presented in the serum sample (Appendix A). We further identified the glycopeptides by using Byonic software and filtered the high-confidence glycopeptides with PEP2D < 0.01. A representative HCD spectra was shown in the intact sialylated biantennary glycopeptide RHEEGHMLN^540^CTCFGQGR (*m*/*z* 1001.40, z = 4+) of fibronectin (FN1) protein from HCC patient serum (Figure 4A). 

The glycopeptide identity was confidently confirmed (Byonic score = 269.1, PEP2D = 0.0001) by the remarkable Y1 ion (*m*/*z* 1146.98, z = 2+), with peptide backbone and one GlcNAc, the Y-series of glycopeptide fragments, and b/y fragment ions from peptide backbone. In the example case, the O-linked sialylated glycopeptide, SLGNVNFTVSAEALES^879^QELCGTEVPSVPEHGR (*m*/*z* 1164.02, z = 4+) of alpha-2-macrogobulin (A2M) protein with intact sialylated tri-sialylateded-bi-antennary glycan, was also detected (Byonic score = 1108.5, PEP2D = 2.13 × 10^−24^, Figure 4B). Based on the criteria (PEP2D < 0.01), a total of 741 glycopeptides were identified, which included 506 sialylated glycopeptides using 200 μg ZIC-cHILC@MNPs with 3 μL of sample injection volume (Figure 4C). By using ZIC-cHILIC@MNPs (200 μg), only 3 μL of HCC serum samples was required for maximum glycoproteomics profiling and was applied in further experiments. We further evaluated the performance of ZIC-cHILIC@MNPs for sensitivity in low amounts (0.2 μL–5 μL) of patient serum. A total of 695 glycopeptides, including a high percentage of 470 sialylated glycopeptides (68%), were enriched from 2 μL HCC serum (Figure 4D). Furthermore, using as low as 0.2 μL of HCC serum, ZIC-cHILIC@MNPs were still able to identify 186 sialylated glycopeptides. Compared to the commonly used amount of serum (15–20μL) [32,49] these results indicated that ZIC-cHILIC@MNPs had excellent profiling sensitivity for low amounts of biofluid samples.

Based on the above results, we further applied this strategy to analyze the glycoproteomic profiles from serum (2 μL) of individual patients with HCC (*n* = 3) and HBV (*n* = 3). Good enrichment specificity (80–91%) was observed in all the patient serums; more than 97% enriched glycopeptides were sialylated glycopeptides (Appendix A). This test result in the serum indicates that ZIC-cHILIC@MNPs possessed excellent specificity for glycopeptide enrichment. A total of 1599–1811 unique glycopeptides from 161 glycoproteins were identified from three HCC patients, in which 1059–1244 were sialylated glycopeptides (Figure 5A and Appendix A). At the same time, a total of 1468–1929 unique glycopeptides corresponded with 147 glycoproteins identified from HBV (*n* = 3) patients, in which 1176–1384 were sialylated glycopeptides (Figure 5A and Appendix A). In addition, identification and classification of N- and O-linked glycopeptides were also conducted. 

Combining patient groups, a total of 3078 and 3163 unique glycopeptides were identified from HCC and HBV patient serum, respectively, in which the majority (88%) were N-glycosylated, whilst 12% were O-glycosylated (Figure 5B). These results indicate that ZIC-cHILIC@MNPs had an excellent enrichment performance, including with low sample amount, high enrichment specificity, and compatibility for both N- and O-linked glycopeptides. Most interestingly, they demonstrated strength for enrichment for sialylated glycopeptides, which have been usually underexplored by existing methods.

Comparing the total of 4383 glycopeptides identified from the HBV and HCC groups, 1858 (42.4%) were commonly presented in the two patient cohorts (Figure 5C). In order to investigate the differential pattern between HCC and HBV patients, quantitation of the abundance of glycopeptides of these six patients was calculated based on the peak area using label-free quantitation with peptide alignment by Proteome Discoverer 2.5 software. The fold change of glycopeptide abundance in the HCC compared to the HBV were evaluated by statistical analysis. The 23 differentially expressed glycopeptides, including 21 sialylated glycopeptides and two high-mannose types of glycopeptides (>2-fold, student *t*-test, *p* < 0.05) were further analyzed by hierarchical clustering (Figure 5D). The results discovered some up-regulated sialylated glycopeptides which have been reported as potential markers in liver cancer from published literature, including KNG1, HPX, TF, and C4 protein in HCC patients [50]. For example, mono-sialylated-tetra-antennary glycan at N241 site of haptoglobin protein shows 3-fold higher in the HCC compared to the HBV group. It has been reported to be overexpressed in HCC compared to non-cancer patients with other liver diseases [49,51]. Therefore, the result further demonstrated the advantages of ZIC-cHILIC@MNPs for quantitative glycopeptide profiling in patient biofluid samples.

## 4. Conclusions

In conclusion, magnetic nanoparticles functionalized with n-dodecylphosphocholine (ZIC-cHILIC@MNPs) were successfully prepared. This novel nanoprobe showed several unique strengths, including a super-fast enrichment process (10 min), high enrichment specificity, and high sensitivity towards low abundance glycopeptides both from standard glycoprotein and serum samples of HCC and HBV patients, likely due to the large active surface area, high hydrophilicity, and good biocompatibility of this nanomaterial. The developed platform integrating the ZIC-cHILIC@MNPs and label-free proteomics quantitation provide a high-performance tool for glycoproteomics profiling of biological samples such as human serum. Using this strategy, we were able to achieve personalized profiles at the depth of thousands of glycopeptides, especially for the large-scale identification of underexplored sialylated glycopeptides. Based on the demonstration of excellent glycopeptide enrichment ability using the real-world samples from HBV and HCC patients, we expect that the ZIC-cHILIC@MNPs would be a generic tool for advancing the glycoproteome analysis, as well as making a significant contribution to the screening of associated biomarkers.

## Figures and Tables

**Figure 1 nanomaterials-11-02159-f001:**
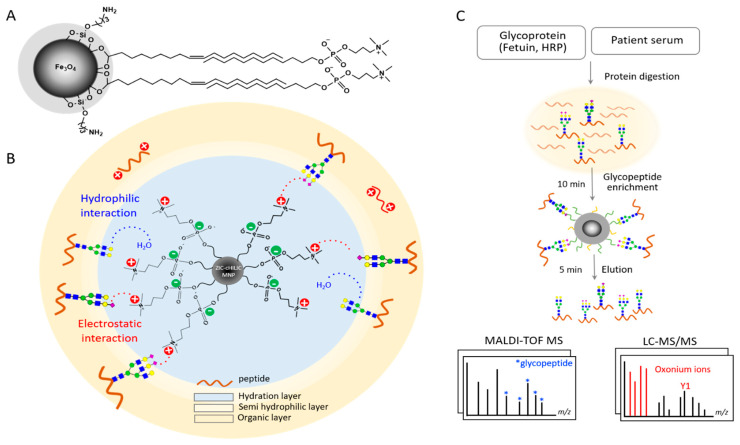
Proposed mechanism and workflow of glycopeptide enrichment using ZIC-cHILIC@MNPs. (**A**) Structure of ZIC-cHILIC@MNPs. (**B**) Proposed mechanism and (**C**) workflow of glycopeptide enrichment by using ZIC-cHILIC@MNPs.

**Figure 2 nanomaterials-11-02159-f002:**
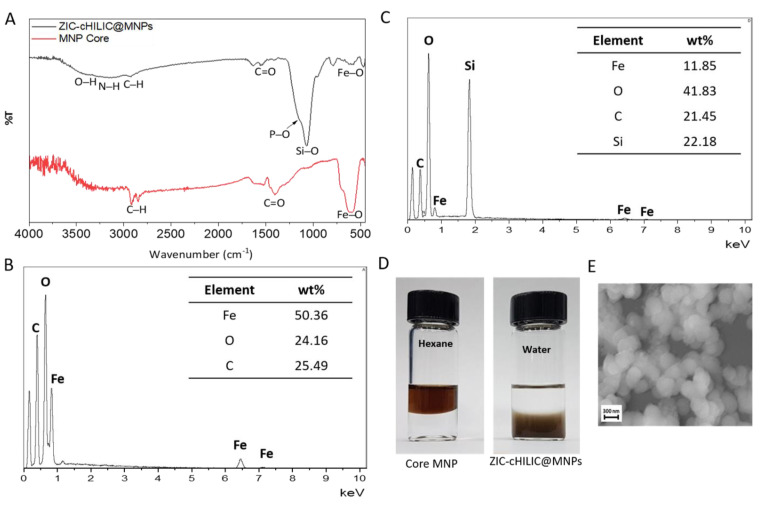
Characterization of ZIC-cHILIC@MNPs. (**A**) FTIR result and, (**B**,**C**) EDX of core MNPs and ZIC-cHILIC@MNPs, (**D**) polarity test of core MNPs and ZIC-cHILIC@MNPs, respectively. (**E**) FE-SEM image of ZIC-cHILIC@MNPs.

**Figure 3 nanomaterials-11-02159-f003:**
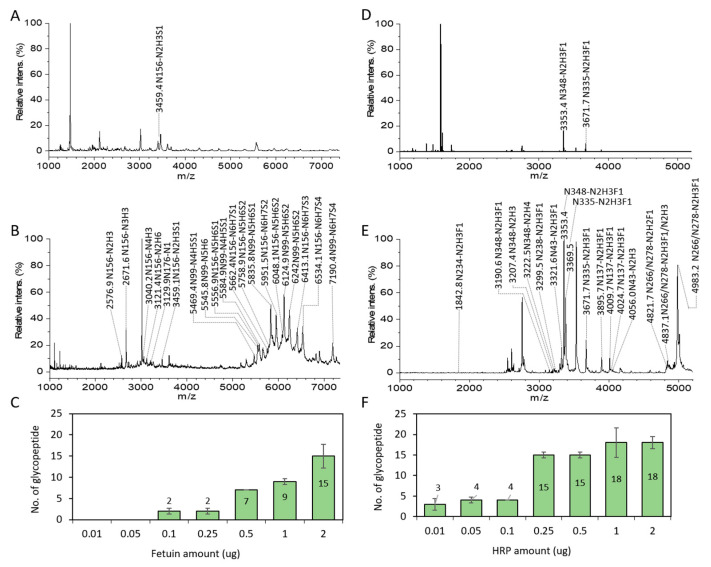
Detection of intact glycopeptides from 2 µg fetuin and HRP by MALDI-TOF mass spectrometry analysis. MALDI-TOF spectra of standard fetuin and HRP before enrichment (**A**,**D**) and after enrichment (**B**,**E**) using ZIC-cHILIC@MNPs, respectively. Number of detected glycopeptide against different amounts of standard fetuin (**C**) and HRP (**F**) using ZIC-cHILIC@MNPs.

**Figure 4 nanomaterials-11-02159-f004:**
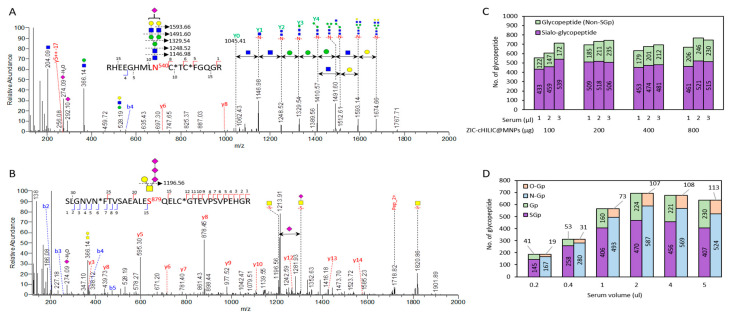
Identification of intact sialylated glycopeptides from HCC serum sample. (**A**) Representative MS/MS spectrum of identified intact N-glycopeptide RHEEGHMLNCTCFGQGR from FN1 protein with mono-sialyl-bi-antennary glycan (*m*/*z* 1001.40, z = 4+, Byonic score = 269.1). (**B**) intact O-glycopeptide SLGNVNFTVSAEALESQELCGTEVPSVPEHGR from A2M (*m*/*z* 1164.02, z = 4+, Byonic score = 1108.5). (**C**) Number of identified glycopeptides for optimization of ZIC-cHILIC@MNPs particle amount, injection sample amount, and (**D**) HCC serum volume.

**Figure 5 nanomaterials-11-02159-f005:**
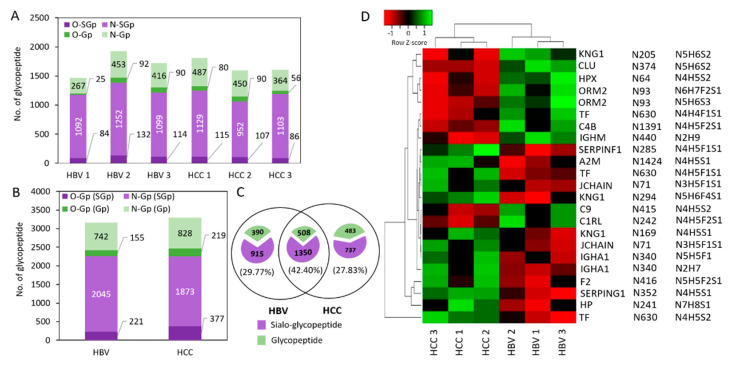
Exploring the differential glycopeptide from individual serum HBV and HCC patients. (**A**) Comparison of unique N-glycopeptide and O-glycopeptide identified from individual HBV and HCC serum and (**B**) two patient cohorts. (**C**) Venn diagram of unique glycopeptide identified from two patient cohorts. (**D**) Hierarchical heatmap of differential glycopeptide abundance using Euclidean distance and complete linkage.

## Data Availability

The data is included in the main text and/or the Appendix A.

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
