# Peer review of "ZIC-cHILIC Functionalized Magnetic Nanoparticle for Rapid and Sensitive Glycopeptide Enrichment from <1 µL Serum"

_nanomaterials, 2021, doi:10.3390/nano11092159_

Round 1
Reviewer 1 Report
The authors report an affinity nanoprobe for the enrichment of glycopeptides based on magnetic nanoparticles functionalized with the zwitterionic n-dodecylphosphocholine. The strategy employs mass spectrometry as a technique for the identification of glycopeptides with high sensitivity allied with the high enrichment specificity provided by the magnetic nanoparticles. This work expands the biosensing and separation applications of magnetic nanoparticles for quantitative glycopeptide profiling, besides affording a valuable tool for glycoproteome analysis. Therefore, the acceptance of this nice work by Nanomaterials is suggested after addressing some issues.
- In the Abstract section, 1st line, clarify to what “their unique structures” refers.
- In the manuscript, the term “zwitterionic” is often written as “zwitter-ionic”. The same format should be used for consistency, probably as “zwitterionic”. The same applies to the term “zwitterion”.
- The advantageous use of magnetic nanoparticles should be highlighted in the abstract.
- In the subsection “2.2 Synthesis of ZIC-cHILIC@MNPs”, 2nd paragraph, specify if in the phrase “Then, 250 μL of tetraethyl orthosilicate and 3 mL of ethyl acetate were added” the reagents are added sequentially or as a mixture.
- Figure S1 should be cited in the text accordingly.
- The quality of figures 3,4 and 5 should be improved.
- In the supplementary information, the colour of the functional groups of ZIC-cHILIC MNP schematic figure does not match the colour of the scheme’s legend.
- The authors performed FT-IR and EDX for the characterization of nanoparticles. However, the results don’t display the contributions from the phosphate group of the n-dodecylphosphocholine. Besides, Figures 2B and 2C correspond to the same result. Why is the contribution from the phosphate not observed? An FT-IR spectrum of each synthesis step product should be included.
- Considering the important role of the electrostatic interactions, as well as the reproducibility of the developed strategy, the authors are suggested to carry out assays of Electrophoretic Light Scattering to measure the zeta potential.
- Regarding the stability of the nanoparticles, did the authors carry out studies of the long-term stability in mild environmental conditions and in the incubation buffer? Besides, did the authors observe aggregation of the nanoparticles during/after the enrichment with glycopeptides? Dynamic Light Scattering could be helpful to clarify these issues.
Reviewer 2 Report
In this research, authors prepared phosphocholine-functionalized magnetic nanoparticles and showed their ability for glycopeptide enrichment. Results of glycopeptide analyses support their conclusion. As the rapid and sensitive analysis of glycopeptide is an important topic, it could be appealing. However, there are some issues to be clarified and revised. Thus, I believe that authors need to revise their manuscript for the publication in Nanomaterials as a research article. My major and minor comments are shown below.
Major comments
- The section 3.1 doesn't have any experimental results and discussion. It should not be described in "results and discussion". Also I don't understand this section title. These are mostly experimental design. Thus, it should be described in the "introduction". Authors should remove this section (3.1) and revise their introduction.
- Authors mentioned that N-dodecylphosphocholine was attached on the MNPs via comprehensive van der Waals interaction. I'm wondering is this really major motive force? How strong is their interaction? How is the stability of N-dodecylphosphocholine on the surface. Also for the surface modification, there must be an extra (excessive) 1,2-hexadecanediol, which could attach to the surface not only by the way of diol attached on NPs but also by the way same as N-dodecylphosphocholine attachment. It could have stronger interaction than N-dodecylphosphocholine does. I think it should be competitive attachment. As authors mentioned, attached phosphocholine is important for this system. However, there are no information about amounts and stability of attached N-dodecylphosphocholine. FT-IR and EDX results don't support their attachment, because they are qualitative and can change by attachment of aminopropyl silane and 1,2-hexadecanediol without N-dodecylphosphocholine. As the amount (or surface coverage) and stability of attached N-dodecylphosphocholine should affect their ability, authors should investigate these.
Minor comments
- Chemical structure in Fig. 1A is incorrect. Authors used 1,2-hexadecanediol for the surface modification of magnetite nanoparticles. It has no unsaturated carbon. Also number of carbon atom and position of OH are also wrong.
- There are many miswriting; for example, "on monolithic nano-LC column for glycopeptide enrichment, for glycopeptide enrichment," (duplication on page 2), "using TCEP final 5 mM of at 37°C" (on page 4), "with 12% are were O-glycosylated" (on page 10), "The develop platform" (on page 11), etc.
- "2" of H2O must be shown as a subscript.
Round 2
Reviewer 1 Report
The authors answered all the questions and comments, which strongly improved the quality of the paper. Therefore, the acceptance of this work by Nanomaterials is suggested.
Reviewer 2 Report
Authors answered all my questions and comments and revised their manuscript. Although I still feel uncertainty on the mechanism from their characterization of ZIC-cHILIC@MNPs, their performance for glycopeptide enrichment is well shown and support their idea. Thus, I suggest to publish this manuscript in Nanomaterials as a research article after minor revision. The points to be revised are shown below.
Minor comments
- There are still some miswriting. Some of them are shown below.
(Page 2) Several studies "have incorporate" the magnetic nanoparticle...
(Page 4) The synthesis of "Fe3O4" magnetic nanoparticle was...
(Page 5) "oC"
Also please check all the "MNP", which must be "MNPs".
- (Page 7) Authors newly added "The peak for P-O stretch of n-dodecylphosphocholine was barely observed...". Then, for better understanding to readers, authors had better to show "P-O" showing its position in Fig.2 A like others such as "Si-O". And also, author had better to indicate peak position for Fe-O, such as "Fe-O (xx-xx cm-1)", in the following sentence in the main text; "The decrease in peak intensity of Fe–O in ZIC-cHILIC MNP curve indicates additional surface coating".
- (Page 7) Authors showed zeta potential of core MNPs and ZIC-cHILIC@MNPs are 1.6 ±20.2 and -22.2±4.8. Although they showed details in the methods, authors had better to show their solvent conditions, such as "core MNPs in hexane and ZIC-cHILIC@MNPs in water", to help readers' understanding. Large deviation for core MNPs is expected to come from a non-polar solvent.
